# Partial Parallelism Plots

Axel Petzold [1,2] 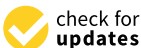

1   Queen Square Institute of Neurology, UCL, London WC1N 3BG, UK; a.petzold@ucl.ac.uk
2   Amsterdam UMC, De Boelelaan 1117, 11081 HX Amsterdam, The Netherlands

**Featured Application:** This article proposes a novel graphical approach for the the assessment of parallelism of biomarker tests that takes into consideration situations where parallelism is partially lacking. The new approach expands on earlier observations and criticism of the limitations of statistical methods included in the guidelines of regulatory authorities. Researchers in the field concur on emphasising the importance of ensuring the accuracy and reliability; a pertinent point which still remains to be addressed. To this purpose, two primary computational approaches are discussed: (a) statistical assessment and (b) visual assessment. Statistical methods, such as regression analysis and parallelism/non-parallelism indexes, offer precision and objectivity, making them suitable for large datasets and high accuracy requirements. They can detect subtle differences in parallelism that may be missed by visual assessment. However, they assume a linear relationship between analyte concentration and assay response, which may not always hold true. Visual assessment relies on interpreting graphs or charts depicting the biomarker–concentration–response relationship. It is intuitive and can quickly identify gross deviations from partial parallelism, making it useful for screening biomarker assays. Visual assessment may detect non-parallelism due to confounding factors that statistical methods might miss. The graphical method proposed here suggests using partial parallelism plots, which visually depict the relationship between biomarker concentration and assay response for each sample. These plots enable the identification of non-parallelism caused by analytical issues or confounding factors. They assist in determining the optimal range of dilutions for each sample and provide a language that is easily understood by researchers, regulatory authorities, and technicians. For regulatory authorities, this document provides valuable insights into the assessment of partial parallelism for biomarker tests. It highlights the need for both statistical and visual assessment methods to evaluate parallelism accurately. The proposed use of partial parallelism plots can aid in visualising and understanding the relationship between biomarker concentration and assay response. By considering these plots during the evaluation of biomarker assays, regulatory authorities can ensure the accuracy, reliability, and suitability of these tests as trial outcome measures and for clinical use.

**Abstract:** Demonstrating parallelism in quantitative laboratory tests is crucial to ensure accurate reporting of data and minimise risks to patients. Regulatory authorities make the demonstration of parallelism before clinical use approval mandate. However, achieving statistical parallelism can be arduous, especially when parallelism is limited to a subrange of the data. To address potential biases and confounds, I propose a simple graphical method, the Partial Parallelism Plot, to demonstrate partial parallelism. The proposed method offers ease of understanding, intuitiveness, and graphical simplicity. It enables the graphical assessment of quantitative data risk when parallelism is lacking within a defined range. As parallelism may not be consistent across the entire analytical range, the plots focus on partial parallelism. The method can readily be programmed into graphical applications for enhanced interactivity. By providing a clear graphical representation, the method allows researchers to ascertain the presence of parallelism in laboratory tests, thus aiding in the validation process for trials and clinical applications.

**Keywords:** parallelism; biomarker; laboratory; test; graphical statistics



## 1. Introduction

In clinical medicine, precise determination of the concentration of a given compound is essential. For instance, in the case of suspected heart attack, the concentration of specific biomarkers must be determined accurately in a blood sample of the patient to support the diagnosis. Inaccurate mathematical calculations based on laboratory measurements may lead to erroneous biomarker concentrations and misdiagnosis. Therefore, mathematical calculations are heavily relied upon in daily clinical and laboratory practices, but it is crucial to ensure that the underlying assumptions of these calculations are satisfied. This report addresses one such assumption, namely parallelism.

Demonstration of parallelism is crucial for the accuracy of any test based on calculating sample concentration from a standard curve [1]. However, it has been noted that there is no widely adopted universal strategy for assessing parallelism in bioassays. Without assurance of parallelism, investigators are unable to calculate reliable estimates for serum antibody concentrations [1]. To address this issue, it has been suggested to visually compare the slope of logistic-log curves, for which a series of excellent examples have been provided. The authors cautioned against purely statistical assessments of parallelism, as the methods of computation are complex, not readily available in software packages, prone to error unless interpreted correctly, and overly sensitive to negligible departures from parallelism when model precision is high. Furthermore, no guidance was provided on how to interpret the data in cases where there is partial non-parallelism, which may make it challenging for users to determine the appropriate course of action. Notwithstanding this constraint, the parallelism plots initially proposed [1] continue to serve as a valuable graphical tool for evaluating parallelism in laboratory tests, and their significance has been acknowledged in subsequent research. According to this authoritative perspective [2], the experimental validation of parallelism remains a challenging and pivotal aspect in the validation of bioanalytical methods to this day, an assertion that was reiterated in a highly influential white paper [3].

Regulatory authorities impose strict requirements for the approval of an assay, including the demonstration of parallelism. As per the latest guidelines by the Food and Drug Administration (FDA) and European Medical Agency (EMA), parallelism is defined as "Parallelism demonstrates that the serially diluted incurred sample response curve is parallel to the calibration curve" [4]. The guideline provides explicit laboratory instructions for conducting the study, involving the dilution of a high-concentration study sample to at least three concentrations with a blank matrix. However, the interpretation of results becomes more ambiguous. The guideline states that the consistency of back-calculated concentrations between samples in a dilution series should not exceed a 30% coefficient of variation (CV). Nevertheless, it is essential to carefully monitor the data, as results meeting this criterion may still indicate trends of non-parallelism. In cases where the sample does not dilute linearly, a predefined procedure for reporting results should be established. In this report, I propose a simple graphical approach for such a procedure, as it may offer greater intuitiveness and be less susceptible to the limitations previously recognised in purely numerical methods [1,2].

The concept of parallelism may appear simple at first glance, but it can be difficult to understand upon further examination. Additionally, numerically driven, statistical representations of parallelism may not be intuitive for individuals without a statistical background. This lack of understanding can be problematic for regulatory authorities and mixed expertise panels tasked with making decisions in laboratory-based research.

## 2. The Range of Accuracy and Effect Size in the Assessment of Laboratory Tests

Experimental evidence indicates a significant impact of the lack of parallelism on the quantification of neurofilaments, a well-established biomarker for neurodegeneration [5,6]. The FDA and the EMA approved the use of two novel drugs based on laboratory results quantifying neurofilaments. A state-of-the-art randomised controlled trial (RCT) demonstrated a reduction in neurofilament blood levels as proof of efficacy for a novel

disease-modifying treatment in multiple sclerosis [7], and another RCT [8] lead to rapid FDA approval for the antisense oligonucleotide tofersen to treat amyotrophic lateral sclerosis. However, neither study considered the possibility of partial non-parallelism of neurofilaments. Although not currently relevant in studies with large effect sizes, such as [7,8], non-parallelism becomes more pertinent in studies with smaller effect sizes, such as those encountered in the large number of trials on Alzheimer's disease which employ biomarkers as an outcome measure.

Accepting that parallelism is a vital factor in the evaluation of laboratory tests for biomarkers, it needs to be acknowledged that parallelism is just one among several other factors influencing test reliability [9]. Pum emphasised that analytical and clinical specificity and sensitivity are additional critical factors [2]. Various biological and technical factors, such as matrix effects, variations in biomarker metabolism, or variations in laboratory test procedures, can also influence the accuracy of laboratory tests for biomarkers. A large international consortium underscored the importance of using high-quality samples [10]. Furthermore, prospective experimental evidence highlighted that the inter-laboratory reproducibility and technician skills are other key factors affecting test outcomes [11].

Assessing the range of accuracy of laboratory tests for biomarkers is a complex task that depends on multiple factors in addition to parallelism. It is crucial to be aware of these factors and to critically evaluate laboratory tests to determine their suitability as diagnostic tools and trial outcome measures in medicine.

## 3. The Definition of Parallelism and Partial Parallelism

The term parallelism, in its simplest definition, describes the relationship between the concentration of an analyte (such as a biomarker) in a sample and the signal produced by the reference standard of the laboratory test used to measure that analyte as earlier introduced [1,2]. When the relationship between concentration and signal is linear, parallelism is said to be present. Hence, the other term used in the literature for parallelism is linearity. This is important because it means that the laboratory test accurately reflects the concentration of the analyte in the sample and, therefore, provides a reliable measurement of the biomarker.

However, if there is non-parallelism (i.e., a non-linear relationship between concentration and signal), the accuracy of the laboratory test may be compromised. This can occur if there is interference from other substances in the sample, or if the laboratory test is not able to accurately detect the analyte, for example, at higher concentrations. This is a frequent problem with biomarker assays requiring use of a non-linear standard curve, as reviewed theoretically in [2] and demonstrated experimentally in [5].

In order to test for linearity, a regression analysis is performed to determine the slope of the line of best fit. The formula for the slope of the line is:

$$slope = \frac{\sum\limits_{i=1}^{n} (x_i - \bar{x}) \times (y_i - \bar{y})}{\sum\limits_{i=1}^{n} (x_i - \bar{x})^2} \tag{1}$$

where

$x_i$ is the concentration of the biomarker in the sample;
$y_i$ is the signal produced by the test used to measure the biomarker;
$n$ is the number of data points;
$\bar{x}$ is the mean concentration of the biomarker;
$\bar{y}$ is the mean signal produced by the laboratory test.

If the slope is not significantly different from 1 (i.e., if $|slope - 1| \leq SE$ where $SE$ is the standard error), then parallelism is present. The values of $x_i$ are the given concentrations (i.e., ng/mL, pg/mL, g/L), and formula (1) uses those values to calculate the slope of the line of best fit, which represents the relationship between the concentration of the analyte

and the signal produced by the laboratory test, as detailed in an entire book chapter [9]. It follows that for biomarker assays with proof of linearity, a parallelism coefficient close to 1 indicates that the patient sample and standard curve have similar slopes:

$$\text{Parallelism coefficient} = \frac{\text{Slope of Patient Dilution}}{\text{Slope of Standard Curve}} \tag{2}$$

In absence of linearity, determination of parallelism was defined for bioassay dilution curves in absence of a standard curve by a logistic-log model in which the signal for the test are optic densities (OD) as:

$$OD = d + \frac{a - d}{1 + (\frac{dilution}{c})^b} \tag{3}$$

where

*a* is the upper asymptote of the curve of the OD at a theoretical infinite concentrations;
*d* is the lower asymptote of the curve of the OD at a theoretical zero concentrations;
*b* is a curvature parameter;
*c* is the symmetry point of the sigmoid.

Linear transformation of the curve is achieved through a logistic function where $OD_{min}$ and $OD_{max}$ correspond to bespoke upper and lower asymptotes as

$$Logit(OD)_{fs} = \log\left(\frac{OD - OD_{min}}{OD_{max} - OD}\right) \tag{4}$$

The formula can be reduced to express a partially specified logit model introduced in [1]:

$$Logit(OD)_{ps} = \log\left(\frac{OD}{OD_{max} - OD}\right) \tag{5}$$

The visualisation of this approach is illustrated in Figure 1. Clearly, none of these curves in Figure 1A meet the criteria for linearity as defined at the onset of this section. Only bespoke logistic-log transformation permits to demonstrate parallelism (Curves 1–4 in Figure 1B) and lack of parallelism (Curve 5 in Figure 1B). The logistic-log transformation is the basis for the statistical analysis of a dilution series intended to facilitate visualisation as intended [1].

$$\text{Relative dilution}_i = 100 \times \left(\frac{\text{actual sample dilution}_i}{\text{maximum dilution in series}}\right) \tag{6}$$

where *i* indicates the dilution step. Finally, Plikaytis et al. used Generalised Linear Models (one-way analysis of covariance) for determination of parallelism.

In summary, parallelism is described as a critical aspect of the analytical test validation in the context of analytical linearity (i.e., regression analysis (Formula (1)), parallelism coefficient (Formula (2))) and non-linearity (i.e., a logistic-log model (Formulas (3)–(5))). The logistic-log transformation ensures that the test can accurately measure the concentration of the biomarker over a range of concentrations [1]. Overall, the choice of method for assessing parallelism depends on the data distribution (i.e., linear or nonlinear).

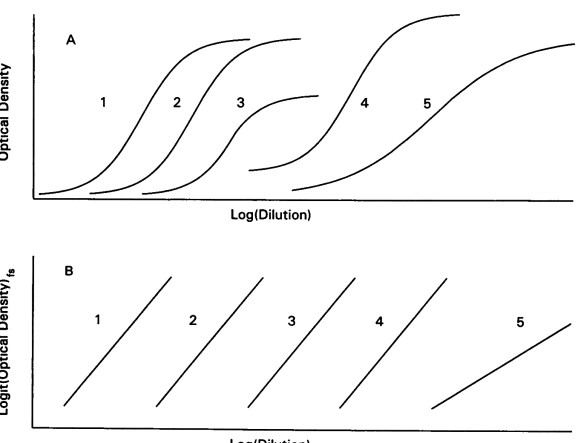

**Figure 1.** Comparison of logistic-log curves and their fully specified logit-log transformed counterparts. (**A**) Lines 1 and 2, logistic-log curves with identical slopes and asymptotes; Lines 3 and 4, logistic-log curves with identical slopes and different asymptotes; Line 5, logistic-log curve with different slope and asymptotes. (**B**) Corresponding straight lines formed by using the fully specified logit-log transformation. [Reproduced with permission from [1]].

## 4. Graphical Presentations of Parallelism

Graphical presentations of parallelism can provide a visual representation of the accuracy and reliability of laboratory tests for biomarkers. Graphical presentations can help healthcare providers and researchers to rapidly (i.e., at a glance) identify potential issues with biomarker tests, such as interference from other substances in the sample or limitations with the analytical sensitivity of the test. They can also be useful for comparing the accuracy of different laboratory tests for the same biomarker.

One common graphical presentation of parallelism is the parallelism plot, which involves plotting the signal produced by the laboratory test on the *y*-axis and the concentration of the biomarker on the *x*-axis. A seminal example from the literature was presented in Figure 1. If the lines are parallel, it suggests that the laboratory test accurately reflects the concentration of the biomarker in the sample and that parallelism is present.

One development in the biomarker field, since the introduction of the logistic-log transformation [1], has been the use of calibrated and quality controlled protein standard curves. Consequently, reported biomarker concentrations are derived from the curve between the symmetry point (c as defined for Formula (3)), but never from the asymptotes. The lower asymptotes (d) indicate non-measurable data. This is either because the detection limit of the assay is insufficient or because there is nothing there to be measured. For the upper asymptote (a), the concentration of the biomarker is too high to be estimated reliably. Extrapolation is not permitted. Sample dilution is required. Taken together parallelism of a biomarker is therefore only determined for

$$(x_i^{std}, y_i^{std}), \in \{1, \ldots, n^{std}\}, \text{where } y \neq a \vee d \tag{7}$$

Table 1 shows the data used for calculation of the graphical presentation of the curves in Figures 2 and 3. The first step of the data transformation used for the graphical presentation of the partial parallelism plots is to adjust the calculated concentration at each dilution step (*i*) as follows:

$$z_i = x_i^{std} \times i \tag{8}$$

This is followed by normalisation of each value of the transformed series to the value for the lowest dilution step (i.e., dilution 1:1, Table 2) as

$$\bar{z}_i = \frac{z_i}{z_1} \tag{9}$$

**Table 1.** Raw data for the doubling dilution curves used for Figures 2 and 3.

| Dilution | Standard | Sample-A | Sample-B |
|---|---|---|---|
| 1:1 | 10 | 8 | 4 |
| 1:2 | 5 | 4 | 6 |
| 1:4 | 2.5 | 2 | 5 |
| 1:8 | 1.25 | 1 | 4 |
| 1:16 | 0.625 | 0.5 | 3 |
| 1:32 | 0.3125 | 0.25 | 1.5 |
| 1:64 | 0.15625 | 0.125 | 0.75 |
| 1:128 | 0.078125 | 0.0625 | 0.375 |
| 1:256 | 0.0390625 | 0.03125 | 0.1875 |
| 1:512 | 0.01953125 | 0.015625 | 0.09375 |

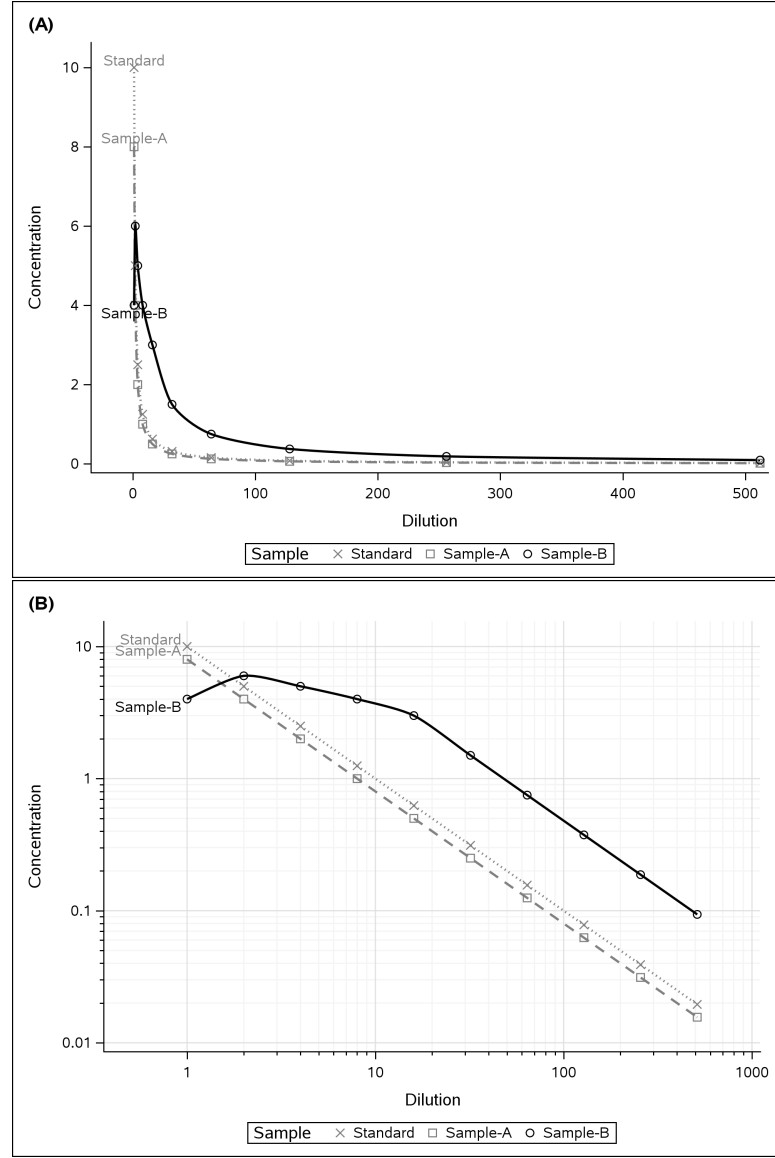

**Figure 2.** Conventional presentations of a doubling dilution curve for demonstration of parallelism between a standard and a sample. This graph illustrates how the concentration of a compound (*y*-axis) decreases with subsequent dilution steps (*x*-axis) either presented as a continuous variable on (**A**), a linear scale as used in [12], and (**B**) on the logarithmic scale derived from Formula (5) [1]. The standard curve (cross, dotted grey line) and dilution curves (Sample-A, open square, dashed grey line; Sample-B, open circle, black line).

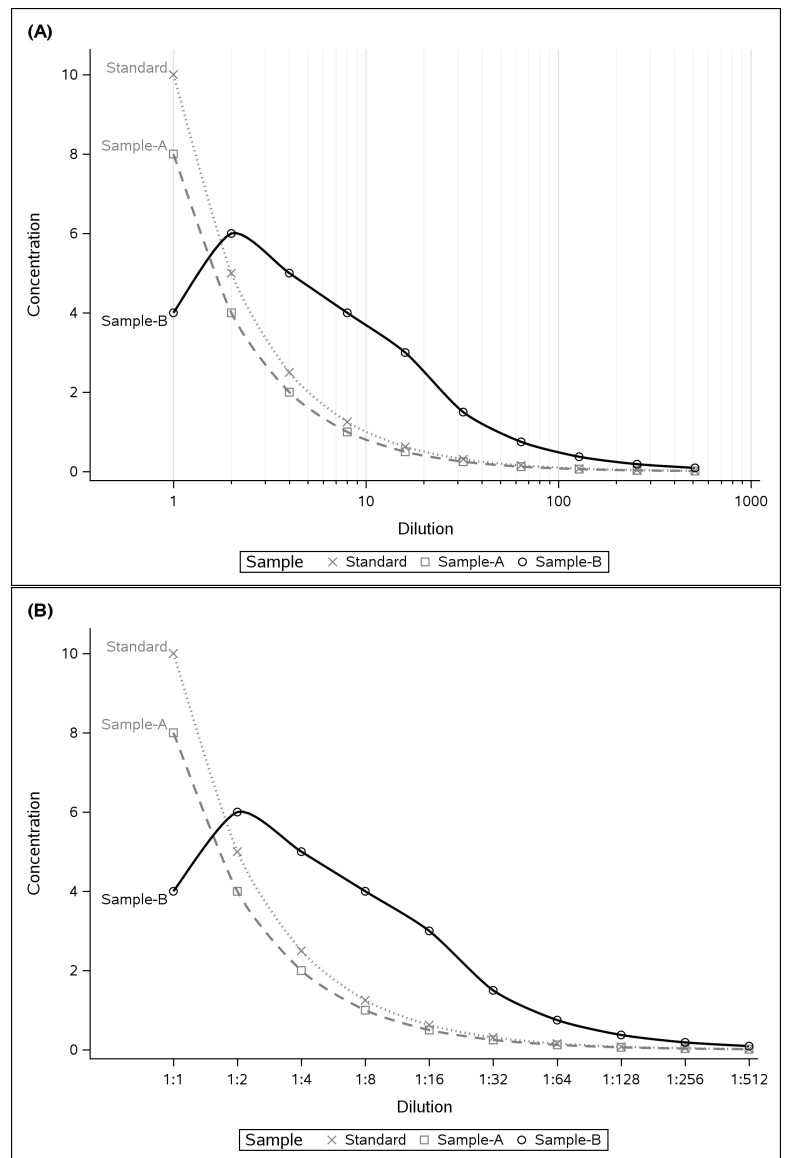

**Figure 3.** Graphical comparison of the partial logistic presentation on the *x*-axis only [1] in (**A**) switched for a categorical variable in (**B**). For any test, the standard (cross, dotted grey line) is used as the main comparator. In this example, the dilution curve for Sample-A (open square, dashed grey line) is parallel to the standard curve. There is a small offset on the *y*-axis between the standard and Sample-A because the starting concentration for Sample-B was less than for the standard. In contrast, the dilution curve for Sample-B (open circle, black line) is not parallel to the standard. For Sample-B, there is an increase in the concentration with the first dilution step. For dilution steps 1:4 to 1:16, the concentration in Sample-B reduces to a lesser degree than for the standard. After Dilution Step 1:32, there is parallelism between Sample-B and the Standard, but this is not clearly visible with this format of graphical presentation.

Overall, graphical presentations of parallelism are an important tool for evaluating the accuracy and reliability of laboratory tests for biomarkers, but there are important practical limitations to their interpretability. The next section will illustrate how this can be overcome in a standardised way which will simplify the interpretation of the graphical presentation.

**Table 2.** Transformed data from Table 1 as needed to develop the partial parallelism plots shown in Figure 4. Abbreviations: Standard = Std, Sample-A = a, Sample-b = b. The horizontal bar above the abbreviation (e.g., $\overline{\text{Std}}$) indicates the normalised values.

| Dil. | Std[1] | A[1] | B[1] | $\overline{\text{Std}}$[2] | $\overline{\text{A}}$[2] | $\overline{\text{B}}$[2] |
|---|---|---|---|---|---|---|
| 1:1 | 10 | 8 | 4 | 1 | 1 | 1 |
| 1:2 | 10 | 8 | 12 | 1 | 1 | 3 |
| 1:4 | 10 | 8 | 20 | 1 | 1 | 5 |
| 1:8 | 10 | 8 | 32 | 1 | 1 | 8 |
| 1:16 | 10 | 8 | 48 | 1 | 1 | 12 |
| 1:32 | 10 | 8 | 48 | 1 | 1 | 12 |
| 1:64 | 10 | 8 | 48 | 1 | 1 | 12 |
| 1:128 | 10 | 8 | 48 | 1 | 1 | 12 |
| 1:256 | 10 | 8 | 48 | 1 | 1 | 12 |
| 1:512 | 10 | 8 | 48 | 1 | 1 | 12 |

[1] Data of concentrations from Table 1 multiplied by dilution step from series (i.e., $5 \times 2 = 10$, $2.5 \times 4 = 10$, etc.) as summarised in Formula (8). [2] Data normalised to concentration at lowest dilution step of the series (i.e., $\frac{10}{10} = 1$, etc.) as summarised in Formula (9).

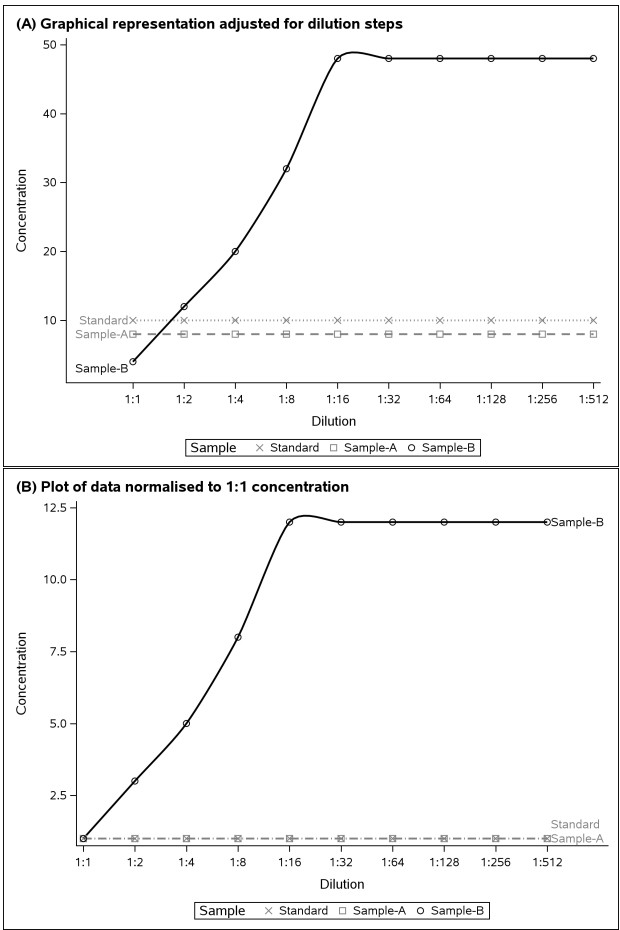

**Figure 4.** (**A**) illustrates that the parallelism between the Standard and Sample-A is visually more intuitive compared to Figure 2. For this presentation, the value of the concentration was corrected by multiplication with the dilution. The offset on the $y$-axis between the Standard and Sample-A is explained by the difference in concentration. This can be a problem for the graphical representation if this difference is very large. Therefore, in example (**B**), all values were normalised to the baseline concentration. Now parallelism between the Standard and Sample-A is illustrated by the overlay of the horizontal lines at the $y$-axis value of one. The consequence of the absence of parallelism for Sample-B in the initial dilution steps leads to an overestimation of a factor of $\approx 12.5$ once parallelism is achieved after a dilution of 1:32.

### 5. Partial Parallelism Plots

In a partial parallelism plot, the laboratory test results are plotted against each other on the *x*- and *y*-axes. A line of unity is then added to the plot to represent perfect parallel agreement between the measurements. The slope of the line, normalised to the first dilution step of the standard curve, is zero with an intercept of one. Therefore, a horizontal reference line at once permits the comparison of the slope of the samples to visually analyse the degree of parallelism for the biomarker in question. If the line of unity (horizontal reference line) is significantly different from the slope of the line of best fit for the sample, it suggests that partial parallelism is absent.

One advantage of partial parallelism plots is that they can be used to assess parallelism between samples over a limited, thought to represent the clinically useful, range of concentrations. This can provide a more practical evaluation of parallelism as relevant for routine healthcare practice.

As a first step towards this goal, Figures 3 and 4 introduce the graphical representation of the line of unity. The result of normalisation for subsequent dilution steps (Formula (7)) is shown graphically for the data from Table 1. In this presentation, there is a similar graphical pattern for the plots in Figure 3A,B. The difference between the two plots can be seen on the *x*-axis. Note that the *x*-axis in Figure 3A is log based. Whilst mathematically correct, this presentation does not make for an easy laboratory, clinical, or health authority-tuned assessment of the biomarker concentration. A much more common notation is the dilution step as used on the categorical scaled *x*-axis in Figure 3B. A limitation of both graphical presentations is that it cannot readily be seen that parallelism between Sample-B and the line of unity is only achieved after a dilution step of 1:32.

The graphical presentation can be improved to better visualise when parallelism is achieved. Figure 4A gives a graphical representation of the same two plots as in Figure 3, adjusted for the dilution steps. For generalisation, the intercept is normalised to one at the baseline in Figure 4B. This graphical presentation is the basis for the development of partial parallelism plots.

The term partial parallelism plot shall be defined as a defined range of biomarker concentrations for which parallelism between sample and standard can be demonstrated. In laboratory practice, parallelism may only be achieved after a certain dilution step because of, for example, a matrix effect (Table 3). Figure 5A illustrates a theoretical situation with a small matrix effect which persists up to a dilution of 1:4 (see vertical reference line). The graphical presentation for a mildly stronger matrix effect persisting up to a dilution of 1:8 is shown in Figure 5B.

Importantly, lack of parallelism can also be present at later stages of the dilution curve (Table 4), for example, because the concentration of the biomarker is below the detection limit of the assay (i.e., *d* in Formula (3)). Figure 6A shows the graphical presentation for lack of parallelism after a dilution step of 1:128. It would be physically impossible to see the developing lack of parallelism with the curves presented in Figure 3. Finally, Figure 6B illustrates the presence of partial parallelism between a dilution of 1:8 to 1:128. At lower or higher dilution steps there is non-parallelism. Again, this pattern cannot be visually extracted from the graphical presentation in Figure 3.

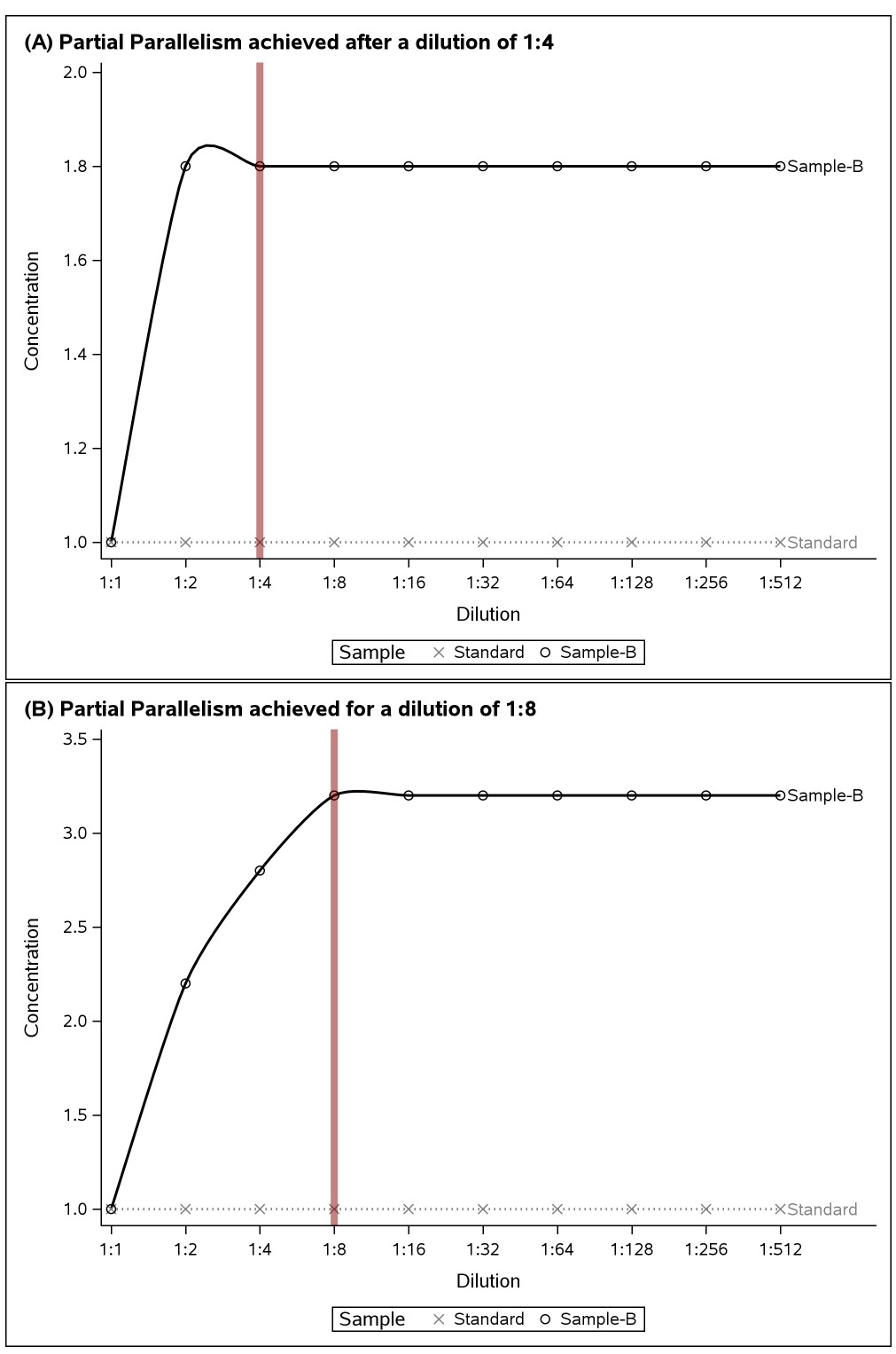

**Figure 5.** These two examples show that parallelism is achieved after (**A**) a dilution of 1:4 and (**B**) a dilution of 1:8. The data are from Table 3.

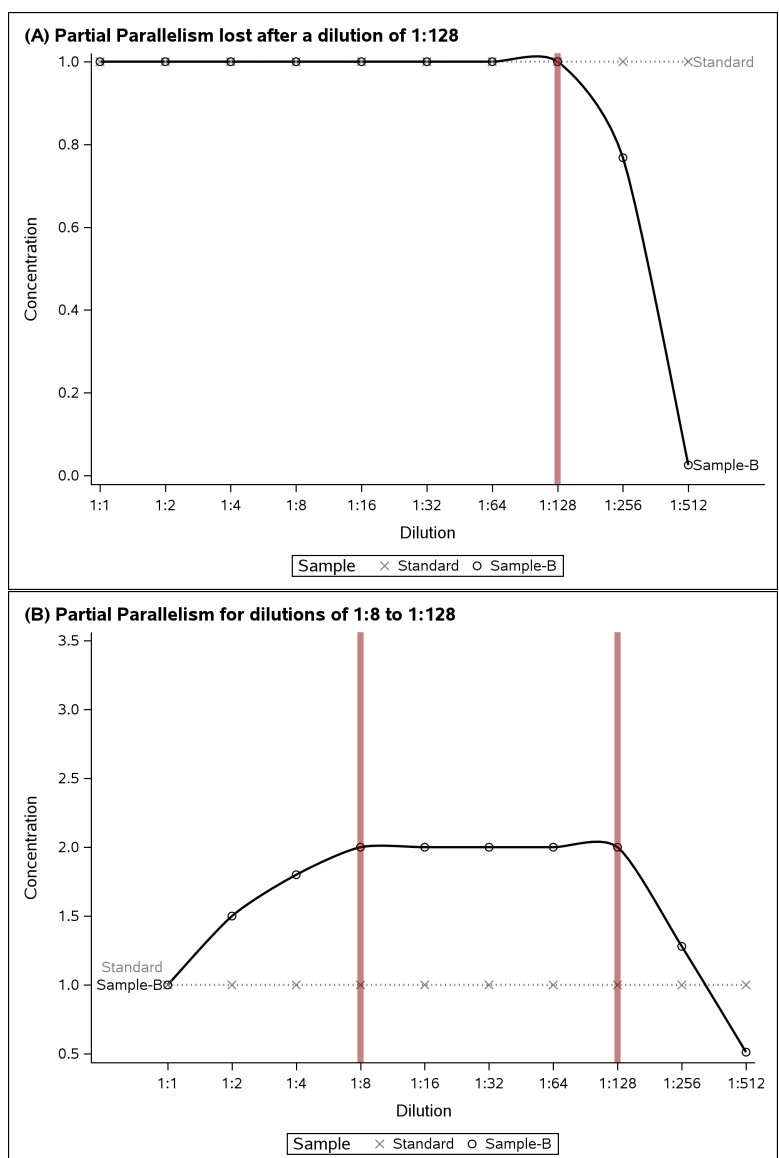

**Figure 6.** These two examples shows that parallelism is lost after (**A**) a dilution of 1:128. Finally, (**B**) illustrates that parallelism was only achieved between a dilution of 1:8 to 1:128. Data are from Table 4.

**Table 3.** Raw and transformed data for the partial parallelism plots shown in Figure 5A,B. Subsequent steps of data transformation are indicated by the superscript in the Table (e.g., $Std^1$, $Std^2$, etc.). The numbers for $\overline{Std^2}$ and $\overline{B^2}$ were used to draw Figure 5.

| Dil. | Std | B | $Std^1$ | $B^1$ | $\overline{Std^2}$ | $\overline{B^2}$ |
|---|---|---|---|---|---|---|
| **(A)** | | | | | | |
| 1:1 | 10 | 9 | 10 | 9 | 1 | 1 |
| 1:2 | 5 | 8.1 | 10 | 16.2 | 1 | 1.8 |
| 1:4 | 2.5 | 4.05 | 10 | 16.2 | 1 | 1.8 |
| 1:8 | 1.25 | 2.025 | 10 | 16.2 | 1 | 1.8 |
| 1:16 | 0.625 | 1.0125 | 10 | 16.2 | 1 | 1.8 |
| 1:32 | 0.3125 | 0.50625 | 10 | 16.2 | 1 | 1.8 |
| 1:64 | 0.15625 | 0.253125 | 10 | 16.2 | 1 | 1.8 |
| 1:128 | 0.078125 | 0.1265625 | 10 | 16.2 | 1 | 1.8 |
| 1:256 | 0.0390625 | 0.06328125 | 10 | 16.2 | 1 | 1.8 |
| 1:512 | 0.01953125 | 0.031640625 | 10 | 16.2 | 1 | 1.8 |

**Table 3.** *Cont.*

| Dil. | Std | B | Std$^1$ | B$^1$ | $\overline{\text{Std}^2}$ | $\overline{\text{B}^2}$ |
|------|------|------|------|------|------|------|
| **(B)** | | | | | | |
| 1:1 | 10 | 1 | 10 | 1 | 1 | 1 |
| 1:2 | 5 | 1.1 | 10 | 2.2 | 1 | 2.2 |
| 1:4 | 2.5 | 0.7 | 10 | 2.8 | 1 | 2.8 |
| 1:8 | 1.25 | 0.4 | 10 | 3.2 | 1 | 3.2 |
| 1:16 | 0.625 | 0.2 | 10 | 3.2 | 1 | 3.2 |
| 1:32 | 0.3125 | 0.1 | 10 | 3.2 | 1 | 3.2 |
| 1:64 | 0.15625 | 0.05 | 10 | 3.2 | 1 | 3.2 |
| 1:128 | 0.078125 | 0.025 | 10 | 3.2 | 1 | 3.2 |
| 1:256 | 0.0390625 | 0.0125 | 10 | 3.2 | 1 | 3.2 |
| 1:512 | 0.01953125 | 0.00625 | 10 | 3.2 | 1 | 3.2 |

**Table 4.** Raw and transformed data for the partial parallelism plots shown in Figure 6A,B. The numbers for $\overline{\text{Std}^2}$ and $\overline{\text{B}^2}$ were used to draw Figure 6.

| Dil. | Std | B | Std$^1$ | B$^1$ | $\overline{\text{Std}^2}$ | $\overline{\text{B}^2}$ |
|------|------|------|------|------|------|------|
| **(A)** | | | | | | |
| 1:1 | 10 | 2 | 10 | 2 | 1 | 1 |
| 1:2 | 5 | 1 | 10 | 2 | 1 | 1 |
| 1:4 | 2.5 | 0.5 | 10 | 2 | 1 | 1 |
| 1:8 | 1.25 | 0.25 | 10 | 2 | 1 | 1 |
| 1:16 | 0.625 | 0.125 | 10 | 2 | 1 | 1 |
| 1:32 | 0.3125 | 0.0625 | 10 | 2 | 1 | 1 |
| 1:64 | 0.15625 | 0.03125 | 10 | 2 | 1 | 1 |
| 1:128 | 0.078125 | 0.015625 | 10 | 2 | 1 | 1 |
| 1:256 | 0.0390625 | 0.006 | 10 | 1.536 | 1 | 0.768 |
| 1:512 | 0.01953125 | 0.0001 | 10 | 0.0512 | 1 | 0.0256 |
| **(B)** | | | | | | |
| 1:1 | 10 | 2 | 10 | 2 | 1 | 1 |
| 1:2 | 5 | 1.5 | 10 | 3 | 1 | 1.5 |
| 1:4 | 2.5 | 0.9 | 10 | 3.6 | 1 | 1.8 |
| 1:8 | 1.25 | 0.5 | 10 | 4 | 1 | 2 |
| 1:16 | 0.625 | 0.25 | 10 | 4 | 1 | 2 |
| 1:32 | 0.3125 | 0.125 | 10 | 4 | 1 | 2 |
| 1:64 | 0.15625 | 0.0625 | 10 | 4 | 1 | 2 |
| 1:128 | 0.078125 | 0.03125 | 10 | 4 | 1 | 2 |
| 1:256 | 0.0390625 | 0.01 | 10 | 2.56 | 1 | 1.28 |
| 1:512 | 0.01953125 | 0.002 | 10 | 1.024 | 1 | 0.512 |

## 6. Examples from the Literature

In a test comparison study [12], parallelism was investigated for allopregnanolone in saliva samples from pregnant women. The sample dilution curves were plotted as in Figure 2A (Figures 1 and 2 in [12]). The conclusion was that the first kit (pg/mL) requires a minimal dilution of 1:5 for an acceptable mean percentage parallelism of 104.3%. The authors accepted that the second kit's test performance met the criteria for parallelism. Using the partial parallelism plot approach, Figure 7A illustrates a lack of parallelism for the first kit. The interpretation of Figure 7A is different to the proposed 1:5 dilution to achieve parallelism [12]. For the second kit, (ng/mL) partial parallelism can be achieved for a dilution range from 1:1 to 1:16, as illustrated by the two vertical reference lines in Figure 7B. At higher dilutions, there is a floor effect of the data suggesting that the test has reached its lower detection limit; hence, the incorrect overestimation of higher concentrations with ever more dilution steps.

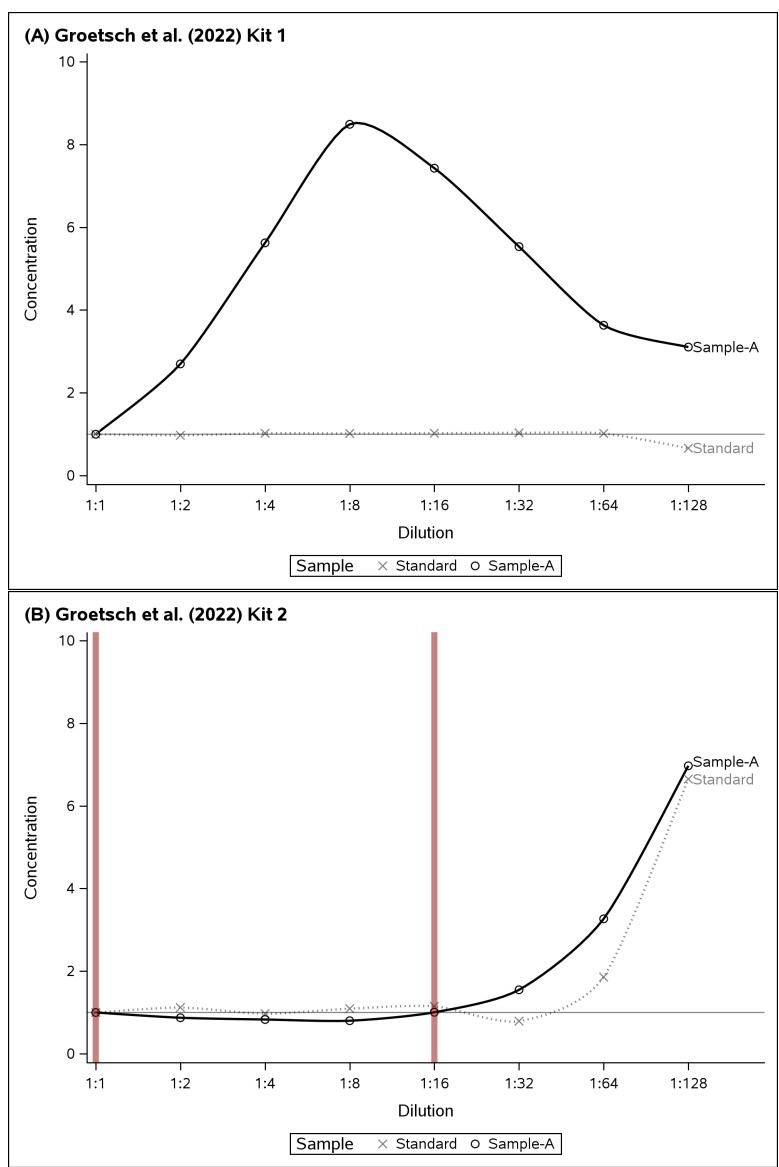

**Figure 7.** Literature example for allopregnanolone quantified by ELISA from saliva samples analysed [12]. (**A**) Lack of parallelism for Kit 1. The maximum error occurs at a dilution step of 1:8 with an ≈8-fold overestimation of the concentration of allopregnanolone (**B**) Partial parallelism between a dilution of 1:1 to 1:16 (red vertical reference lines). In this example, a horizontal black reference line is given at y = 1 which illustrates that there is also a problem with the standard in (**B**), most likely diluted beyond the analytical detection limit of the assay.

There are situations where it is desirable to quantify the same substance from different types of samples. Consequently, an Enzyme-Linked Immunosorbent Assay (ELISA) was developed for measurement of luteinizing hormone (LH) from whole blood, serum, cell extracts, cell culture medium, and pituitary gland extracts [13]. Averaged LH data on the parallelism experiments for these five different sample sources were provided in Tables 6 to 10. Based on these data, Figure 8 shows good partial parallelism for a dilution range from 1:1 to 1:4. After that, near perfect parallelism appears to be lost. The conclusion could be that the concentrations of LH cannot anymore be calculated reliably for comparison from different sources. But the deviation from one on the *y*-axis are only minimal (≤0.2 units). Therefore, in this example, partial parallelism persists up to a dilution of 1:32. After a dilution step of 1:64 the detection limit of the assay is reached for all sample sources.

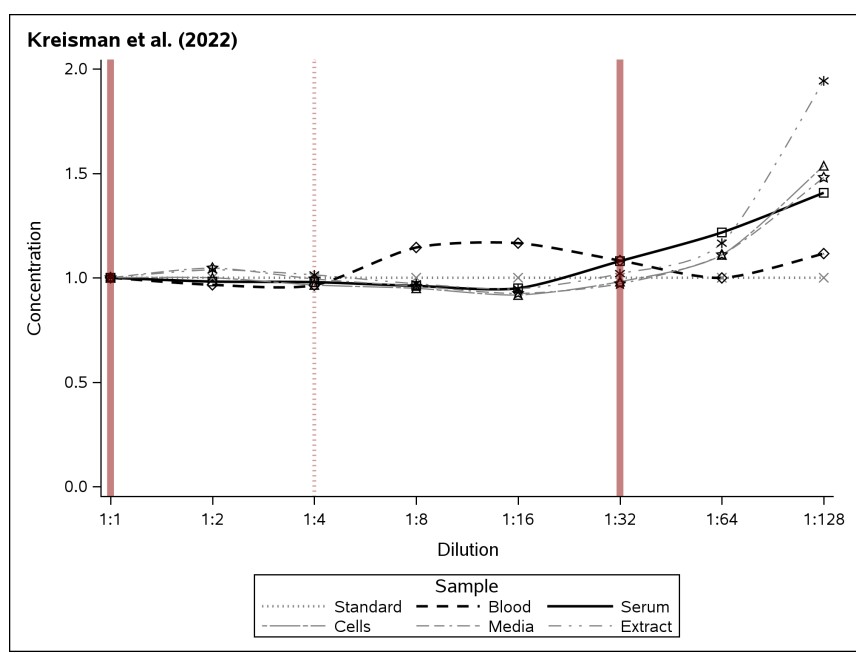

**Figure 8.** Literature example for luteinizing hormone quantified by ELISA from different sources [13]. At first glance, near perfect parallelism weakens after a dilution step of 1:4 (dotted vertical reference line). The error for partial parallelism is, however, minimal (≈0.2 units). Therefore, partial parallelism can be accepted for a dilution range of 1:1 to 1:32 (closed red vertical reference lines).

Lack of parallelism has also been reported explicitly [14]. These authors clarify in the abstract poor "dilution linearity" attributed to "presence of a matrix effect and/or different immunoreactivity of the antibodies to the recombinant standard and the endogenous analyte". The partial parallelism plot in Figure 9 is based on the raw data provided in Table 2 in [14]. Consistent with the author's conclusion, this graph convincingly shows absence of parallelism.

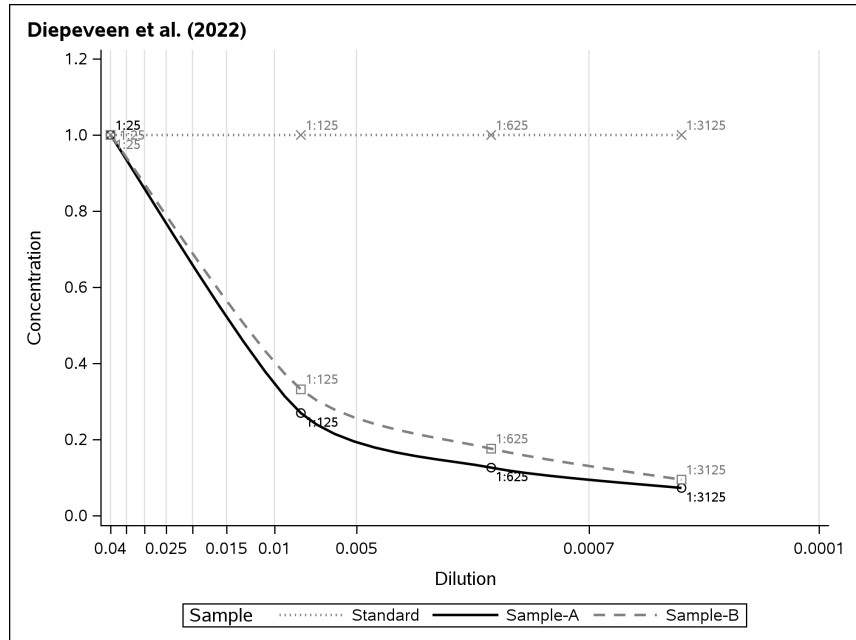

**Figure 9.** Literature example for reported lack of parallelism for erythroferrone quantified by ELISA from human serum samples [14]. Note that in this example an inverse logarithmic scale was used compared to what was presented in Figure 2B. This choice was based on the uneven dilution steps reported for this experiment. For clarity, each data point was labelled with the corresponding dilution step.

Finally, an example for perfect partial parallelism is shown in Figure 10. This example is based on an ELISA for quantification of human insulin (uU/mL) for a dilution range of 1:1 to 1:8 [15]. The data for the partial parallelism plot were taken from Table 4 in [15], which also details that Samples A to D were based on plasma samples with exogenous insulin added (dashed lines in Figure 10) and high endogenous insulin (dotted lines in Figure 10). Note that the range of the *y*-axis presented in Figure 10 is very narrow at only 1 uU/mL insulin (range 0.95 to 1.05 uU/mL). Both spiked (samples with exogenous insulin added) and native samples are perfectly parallel to the standard (solid horizontal reference line at y = 1).

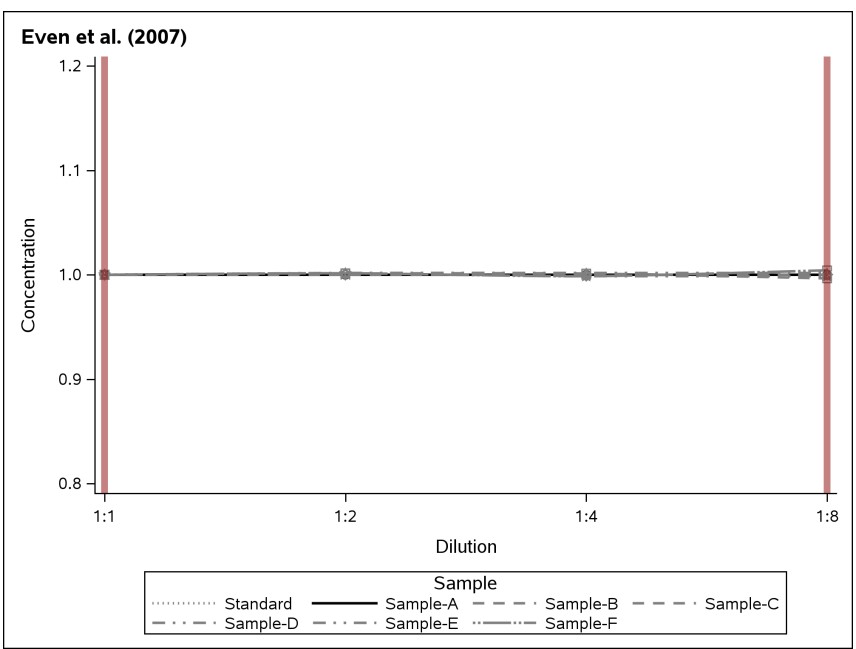

**Figure 10.** Literature example for perfect parallelism for insulin quantified by ELISA from human plasma samples [15]. The error for partial parallelism is negligible (<0.01 units).

Taken together, partial parallelism plots are a useful graphical method for evaluating the accuracy of calculating the biomarker concentration from a sample based on a biomarker standard curve for a defined range of concentrations. Therefore they can provide a more practical evaluation, which is also easy to understand, and can help identify potential sources of non-parallelism.

## 7. Relationships between Confounds and Parallelism

The relationship between confounds and parallelism in laboratory tests is an important topic, as confounds can have a significant impact on the accuracy and reliability of laboratory tests. By definition, confounds are variables that can affect the results of laboratory tests but are not directly related to the biomarker being measured. It has been noted that frequent examples of confounds include chemical stability of the biomarker, repeated freeze–thaw cycles, gender, height, weight, renal function, medication use, and co-morbidities such as diabetes mellitus [9].

Protein biomarker studies have shown that the presence of confounds, including sample preparation and storage, can impact the degree of parallelism between laboratory tests as earlier stated [5,10]. The relationship between confounds and parallelism can be expressed mathematically using regression equations as

$$y = \beta_0 + \beta_1 x + \epsilon \tag{10}$$

where $y$ is the signal produced by the laboratory test, $x$ is the concentration of the biomarker being measured, $\beta_0$ is the intercept, $\beta_1$ is the slope, and $\epsilon$ is the error term. Consequently, confounds can be added to equation (10) as additional independent variables:

$$y = \beta_0 + \beta_1 x + \beta_2 c + \epsilon \tag{11}$$

where $c$ represents a confounding variable. The impact of the confound on the parallelism between laboratory tests can be assessed by comparing the slopes of the regression lines with and without the confound. The need for testing this has been highlighted in a recent white paper [3].

It is important for researchers and healthcare providers to be aware of the potential impact of confounds ($c$) on the accuracy and reliability of the biomarker tests. Laboratories should take steps to minimise the impact of confounds, including controlling for them in statistical analyses or by stratification of the analyses by confounding variables.

Overall, the need for research studies to include testing for confounds for their relationship with the degree of parallelism in biomarker tests has been recognised, but not yet been implemented systematically in the literature.

## 8. Discussion

The practical advantages of partial parallelism plots for biomarker tests has been illustrated statistically and graphically. Application of partial parallelism plots to real biomarker data has revealed the strength of the approach compared to alternatives which were reviewed and discussed with regard to their historical development. The assessment of partial parallelism is an essential step in the validation of laboratory tests, as it determines whether the assay produces accurate and reliable results. There are two primary methods for statistical and graphical assessment. Each method has its own benefits and limitations.

Statistical assessment, such as regression analysis and the calculation of parallelism and non-parallelism indexes, provides a quantitative measure of the degree of parallelism between two or more samples as discussed [1,2]. These methods are precise and objective, making them ideal for the assessment of large datasets or when a high level of accuracy is required. Additionally, statistical methods can detect subtle differences in parallelism that may be missed by graphical assessment.

However, statistical methods also have limitations. They assume that the assay follows a linear relationship between the concentration of the analyte and the response of the assay. This may not always be the case, as assays may exhibit non-linear responses at high or low concentrations. For example, the FDA and EMA guidelines state that "Parallelism is a performance characteristic that can detect potential matrix effects." [4]. A limitation of this definition is that it does not consider (i) the possibility of compound aggregate release or modifiable epitope masking in immunoassays [5]. Additionally, it was highlighted that statistical methods may not detect non-parallelism due to confounding factors, such as matrix effects or interference by endogenous substances, the CV, and the critical difference [16]. Statistical methods are also not necessarily easily comprehensible to many of the parties involved in appraisal of a biomarker test.

Visual assessment, on the other hand, relies on the interpretation of graphs or charts that depict the relationship between the concentration of the biomarker and the response of the assay. This method is very intuitive and can quickly identify gross deviations from partial parallelism, making it useful for screening biomarker assays for technicians, lay people, and regulatory authorities. Additionally, graphical assessment may detect non-parallelism due to confounding factors that are not detected by statistical methods. This includes, for example, a drop in the analytical sensitivity which affects a biomarker and test standard curve similarly (see Figure 7B). There are many chemical and biological reason to the lack of parallelism [17]. The major contributors to non-parallelism are related to interference or a mismatch with the capture antibody (or surface), the detection antibody, the surrogate reference material, the endogenous analyte, and specific and non-specific interactions. For optimal graphical presentation of the concentration range where parallelism applies in a

test, a "raw signal" approach was proposed which includes a four parameter logistic regression curve fitting. The "raw signal" approach is similar to the Figure 3B. Present proposal of partial parallelism plots, as presented in Figures 5–9, should be interpreted as a further simplification of the "raw signal" approach. Individual researchers from all backgrounds and regulatory authorities may also find an advantage in the simplified pattern recognition of the partial parallelism plots. Importantly, both approaches emphasise that parallelism does not need to extend over the entire analytical range of a given test.

However, graphical assessment also has limitations. It is subjective and may vary depending on the experience and expertise of the assessor. Additionally, graphical assessment may not detect subtle deviations from partial parallelism that may affect the accuracy and reliability of the assay. One such factor relates to confounds and was expressed as the CV. In such a situation where the graph may not be clear cut, the visual approach can be improved by showing the parameters obtained from the fits, including the R-square values. Another improvement is the option to zoom into specific regions of the partial parallelism plots. One example was presented in Figure 8. After zooming in, it becomes visibly clearer that the data distribution is more random for the dilution steps 1:2 to 1:16 than for the following dilution steps which clearly demonstrate deviation from parallelism, even if the CV initially improves. Providing this level of interactivity will be a valuable improvement of the method for digital applications making use of proposed partial parallelism plots.

Taken together, both statistical and graphical assessment methods have their own benefits and limitations in the assessment of partial parallelism for biomarker tests. A combination of both methods may provide a comprehensive assessment of the degree of partial parallelism and the presence of non-parallelism due to analytical issues or confounding factors.

## 9. Conclusions

In conclusion, the introduction of partial parallelism plots as a tool for assessing parallelism in biomarker tests holds great promise. These plots offer a clear visualisation of the relationship between biomarker concentration and assay response for each sample, enabling the identification of non-parallelism arising from analytical challenges or confounding factors. Emphasising the importance of determining the optimal range of dilutions for each sample, these plots provide a language that is easily interpretable, ultimately leading to the attainment of accurate and reliable results. As such, incorporating partial parallelism plots into the validation process of quantitative laboratory tests is an essential step to ensure their appropriateness for clinical medicine, bolstering confidence in their utility.

**Supplementary Materials:** The following supporting information can be downloaded at https://www.mdpi.com/article/10.3390/app14020602/s1, Supplementary data are provided in form of an Excel sheet with two tabs for raw data and partial parallelism plot calculations. Table S1: Raw Data for Real-Life Examples: The first tab of this Excel sheet provides comprehensive details of the raw data used to generate the Figures based on real-life examples. Each row corresponds to an individual data point, and the first author's name is referenced for the source of the data. The columns are structured as follows: (a) Dilution Steps (String Variable): This column represents the dilution steps used in the experiments. (b) Dilution Steps (Numeric Variable): This column provides the numeric representation of the dilution steps. (c) Sample Description: This column provides information about the samples used in the experiments. (d) Numeric Values for *Y*-axis (Column 1): The first column containing numeric values used for the *Y*-axis in the Figures. (e) Numeric Values for *Y*-axis (Column 2—Optional): An optional second column containing additional numeric values for the *Y*-axis. Following the raw data presentation, each example is followed by a section detailing the conversion of the *Y*-axis values into values suitable for the partial parallelism plots (PPP plot calculations) presented in this paper. Table S2: Compact Format for PPP Plot Calculations: The second tab of the Excel sheet contains the data organised in a concise 5-column format, specifically designed for easy export into a comma-separated file (.csv). This format is suitable for generating graphical representations used in the present article. The columns are arranged as follows: (a) Author's first name and year of publication. (b) Dilution

Steps (String Variable): This column represents the dilution steps used in the experiments. (c) Dilution Steps (Numeric Variable): This column provides the numeric representation of the dilution steps. (d) Sample Description: This column offers a brief description of the samples. (e) Numeric Values for *Y*-axis (Column 1): The first column containing numeric values used for the *Y*-axis in the Figures. (f) Numeric Values for *Y*-axis (Column 2—Optional): An optional second column containing additional numeric values for the *Y*-axis. The section containing PPP plot calculations serves as a template for readers to conduct their own calculations. It is crucial to verify the accuracy of the dilution steps and ensure that the reference for the "normalised" fields remains unchanged during the process. By presenting the raw data and providing a user-friendly template for PPP plot calculations, this Excel sheet aims to enhance reproducibility and facilitate further research in the field.

**Funding:** This research received no external funding.

**Institutional Review Board Statement:** Not applicable.

**Informed Consent Statement:** Not applicable.

**Data Availability Statement:** All data are provided in the Supplementary Materials.

**Acknowledgments:** Maria Grötsch is gratefully acknowledged for the sharing of raw data. The National Institute for Health Research (NIHR) Biomedical Research Centre based at Moorfields Eye Hospital NHS Foundation Trust and UCL Institute of Ophthalmology are also gratefully acknowledged. The views expressed are those of the author and not necessarily those of the NHS, the NIHR, or the Department of Health.

**Conflicts of Interest:** The authors declare no conflict of interest.

## Abbreviations

The following abbreviations are used in this manuscript:

| | |
|---|---|
| EMA | European Medical Agency |
| ELISA | Enzyme-Linked Immunosorbent Assay |
| FDA | Food and Drug Adminsitration |
| GLM | General Linear Models |
| LH | luteinizing hormone |
| OD | Optical Density |
| SE | Standard Error |

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
