# Peer review of "Partial Parallelism Plots"

_applsci, doi:10.3390/app14020602_

Round 1

Reviewer 1 Report

Comments and Suggestions for Authors

In the paper, the author proposes new visualization tool to detect parallelism or lack of within a sample. The manuscript is clear, and the figures are easy to follow. 

There are no major concerns. 

A few spelling errors are detected while reading the manuscript. For example, "linerisation" is found. Please do a spell check. 

Besides,

  1. What is the main question addressed by the research? The main question is to find another visualization method that is more intuitive, sensitive and informative to practitioners so that they can determine whether a sample meets the criteria for parallelism.
  2. Do you consider the topic original or relevant in the field? Does it address a specific gap in the field? It does address a specific gap as the current way of visualization may not be sensitive for clinicians to determine parallelism.
  3. What does it add to the subject area compared with other published material? It provides a novel way of visualization.
  4. What specific improvements should the authors consider regarding the methodology? What further controls should be considered? One improvement is to show the parameters obtained from the fits and their R-square values. This will help in differentiating the cases where the graph may not be clear cut. Another improvement is to provide interactivity such that the user can zoom into specific regions of the plots. The authors can add this to the discussion. In the clinical setting, when the visualization is employed in a computerized system, it would be great to allow users to have certain interactivity like selecting points, zooming in, and changing the order of the plots (this is particularly important when parts of plots may be overlapping).
  5. Are the conclusions consistent with the evidence and arguments presented and do they address the main question posed? The conclusions are consistent with the evidence and arguments presented. The examples given by the authors about the different cases of parallelism lost are different dilutions is helpful for understanding how to use the visualization.
  6. Are the references appropriate? Yes.
  7. Please include any additional comments on the tables and figures. The tables and figures are fine.

Comments on the Quality of English Language

The English is good. 

Author Response

Referee #1

In the paper, the author proposes new visualization tool to detect parallelism or lack of within a sample. The manuscript is clear, and the figures are easy to follow. 

There are no major concerns. 

A few spelling errors are detected while reading the manuscript. For example, "linerisation" is found. Please do a spell check. 

A: rephrased as “linear transformation.”

Besides,

1. What is the main question addressed by the research? The main question is to find another visualization method that is more intuitive, sensitive and informative to practitioners so that they can determine whether a sample meets the criteria for parallelism.

2. Do you consider the topic original or relevant in the field? Does it address a specific gap in the field? It does address a specific gap as the current way of visualization may not be sensitive for clinicians to determine parallelism.

3. What does it add to the subject area compared with other published material? It provides a novel way of visualization.

4. What specific improvements should the authors consider regarding the methodology? What further controls should be considered?
One improvement is to show the parameters obtained from the fits and their R-square values. This will help in differentiating the cases where the graph may not be clear cut. Another improvement is to provide interactivity such that the user can zoom into specific regions of the plots. The authors can add this to the discussion.

A:Thank you for this suggestion. The following paragraph was added to the limitation section of the method in the discussion:

“In such a situation where the graph may not be clear cut, the visual approach can be improved by showing the parameters obtained from the fits, including the R-square values. Another improvement is the option to zoom into specific regions of the partial parallelism plots. One example was presented in Figure 8. After zooming in it becomes visibly clearer that the data distribution is more random for the dilution steps 1:2 to 1:16, than for the following dilution steps which clearly demonstrate deviation from parallelism, even if the standard deviation initially improves. Providing this level of interactivity will be a valuable improvement of the method for digital applications making use of proposed partial parallelism plots.”

In the clinical setting, when the visualization is employed in a computerized system, it would be great to allow users to have certain interactivity like selecting points, zooming in, and changing the order of the plots (this is particularly important when parts of plots may be overlapping).

A: added to the discussion.

5. Are the conclusions consistent with the evidence and arguments presented and do they address the main question posed? The conclusions are consistent with the evidence and arguments presented. The examples given by the authors about the different cases of parallelism lost are different dilutions is helpful for understanding how to use the visualization.

6. Are the references appropriate? Yes.

7. Please include any additional comments on the tables and figures. The tables and figures are fine.

Reviewer 2 Report

Comments and Suggestions for Authors

This appears to be an important paper which helps in a more practical approach to the issue of parallelism in laboratory medicine. The manuscript details statistical calculations for linearity and parallelism, which may be presented as supplementary material, and is not easy to follow in its present form, given the confusing distribution of figures, text and tables throughout the presentation. However, the message – along with its limitations – is clearly supported by the data.

I have only a few (minor) comments to the Author.

1)  There is not much of a difference between paragraphs 3 (lines 72-85) and 4 (lines 86-100) of the Introduction

2)  Lines 111-12: the predicate is missing

3)  Line 133: typo

4)  Line 177: typo

5)  Lines 186-88: the predicate is missing

6)  Line 210: too many ‘only’

7)  The sequence of text, tables and figures from page 7 to 9 is confusing the reader, and it should be rearranged

8)  Lines 237-8: rephrase the sentence

9)  Lines 245-6: rephrase the sentence

10)  Line 269: typo

11)  Line 337: reviewed AND discussed

Comments on the Quality of English Language

Some changes are required as detailed in the Suggestions for Author

Author Response

Referee #2

This appears to be an important paper which helps in a more practical approach to the issue of parallelism in laboratory medicine. The manuscript details statistical calculations for linearity and parallelism, which may be presented as supplementary material, and is not easy to follow in its present form, given the confusing distribution of figures, text and tables throughout the presentation. However, the message – along with its limitations – is clearly supported by the data.

I have only a few (minor) comments to the Author.

1) There is not much of a difference between paragraphs 3 (lines 72-85) and 4 (lines 86-100) of the Introduction

A: Thank you for pointing out the redundancy which helped to shorten the text of the Introduction.

2) Lines 111-12: the predicate is missing

A: corrected and sentence shortened to: “Experimental evidence indicates a significant impact of the lack of parallelism on the quantification of neurofilaments, a well-established biomarker for

neurodegeneration [8,9].”

3) Line 133: typo

A: corrected.

4) Line 177: typo

A: corrected.

5) Lines 186-88: the predicate is missing

A: corrected and sentence shortened.

6) Line 210: too many ‘only’

A: corrected.

7) The sequence of text, tables and figures from page 7 to 9 is confusing the reader, and it should be rearranged

A: Table 2 has now been re-arranged directly behind Table 1. The Table 2 legend did indeed start by referring to Table 1 and this may be a less confusion location in the manuscript to place it. This will also make it easier to compare the numericals in both tables.

All Figures now appear after these two tables.

8) Lines 237-8: rephrase the sentence

A: corrected.

9) Lines 245-6: rephrase the sentence

A: done

10) Line 269: typo

A: corrected.

11) Line 337: reviewed AND discussed

A: corrected.

Reviewer 3 Report

Comments and Suggestions for Authors

Dear Editor,

Dear Author,

I appreciate the opportunity to review this article and the efforts that the author made for this research.

I find this article elegantly and careful written over the subject of parallelism plots and their role in determining specific concentration for medical biomarkers tests. I appreciate also that the author shared the raw data as supplementary materials attached to this research. I would recommend some minor revision for further processing.

Minor comments:

Please check that sometimes the “Figure” reference in text begins with a capital letter and sometimes you used “figure” without capital, like in the line 212, 234, 237, 238, 241, 245, 247, 255, 260, 263, 298, 299, 371, 372.

The same observation for table reference in the text: in the line 236 is without capital letter. Please choose the appropriate style and use it throughout the entire article.

Please check also the line 245 where you referred the Figure 4 two times in the same phrase in comparison. Maybe you intended a different meaning.

I recommend also explaining all the abbreviations when you use them for the first time like ELISA (Enzyme-Linked Immunosorbent Assay) in the line 278

Thank you for your efforts and good luck further!

Author Response

Referee #3

Dear Editor,

*****

I appreciate the opportunity to review this article and the efforts that the author made for this research.

I find this article elegantly and careful written over the subject of parallelism plots and their role in determining specific concentration for medical biomarkers tests. I appreciate also that the author shared the raw data as supplementary materials attached to this research. I would recommend some minor revision for further processing.

Minor comments:

Please check that sometimes the “Figure” reference in text begins with a capital letter and sometimes you used “figure” without capital, like in the line 212, 234, 237, 238, 241, 245, 247, 255, 260, 263, 298, 299, 371, 372.

A: Figures are now capitalized for the first letter uniformly throughout the manuscript.

The same observation for table reference in the text: in the line 236 is without capital letter. Please choose the appropriate style and use it throughout the entire article.

A: Tables are now capitalized uniformly.

Please check also the line 245 where you referred the Figure 4 two times in the same phrase in comparison. Maybe you intended a different meaning.

A: Thank you for spotting, this was a typo. Figure 3 and 4 were confused. This has been corrected.

I recommend also explaining all the abbreviations when you use them for the first time like ELISA (Enzyme-Linked Immunosorbent Assay) in the line 278

A: Abbreviations ELISA and GLM are now spelled out at first use.

Thank you for your efforts and good luck further!